# The Conserved *Herpesviridae* Protein Kinase (CHPK) of *Gallid alphaherpesvirus* 3 (GaHV3) Is Required for Horizontal Spread and Natural Infection in Chickens

**DOI:** 10.3390/v14030586

**Published:** 2022-03-12

**Authors:** Andrea Krieter, Huai Xu, Haji Akbar, Taejoong Kim, Keith William Jarosinski

**Affiliations:** 1Department of Pathobiology, College of Veterinary Medicine, University of Illinois at Urbana-Champaign, Urbana, IL 61802, USA; krieter2@illinois.edu (A.K.); huaixu2@illinois.edu (H.X.); akbar3@illinois.edu (H.A.); 2US National Poultry Research Center, United States Department of Agriculture, Agricultural Research Service, Athens, GA 30605, USA; taejoong.kim@usda.gov

**Keywords:** herpesvirus, conserved herpesvirus protein kinase, horizontal spread, Marek’s disease vaccine, chicken

## Abstract

We have formerly identified the conserved herpesvirus protein kinase (CHPK) as essential for horizontal transmission of Marek’s disease virus (MDV). Thus far, it has been confirmed that the mutation of the invariant lysine (K) of CHPKs abrogates kinase activity and that CHPK activity is required for MDV horizontal transmission. Since CHPK is conserved among all members of the *Herpesviridae*, we hypothesized that CHPK, and specifically its kinase activity, is important for the horizontal transmission of other herpesviruses. To test this hypothesis, we utilized our experimental and natural infection model in chickens with MD vaccine strain 301B/1 of *Gallid alphaherpesvirus* 3 (GaHV3). First, we mutated the invariant lysine (K) 157 of 301B/1 CHPK to alanine (A) and determined whether it was required for horizontal transmission. To confirm the requirement of 301B/1 CHPK activity for transmission, a rescued virus was generated in which the A157 was changed back to a K (A157K). Despite both the CHPK mutant (K157A) and rescuant (A157K) viruses having replication defects in vivo, only the CHPK mutant (K157A) was unable to spread to contact chickens, while both wild-type and rescuant (A157K) viruses transmitted efficiently, confirming the importance of CHPK activity for horizontal spread. The data confirm that CHPK is required for GaHV3 transmission and suggest that the requirement of avian CHPKs for natural infection is conserved.

## 1. Introduction

The conserved *Herpesviridae* protein kinase (CHPK), a serine/threonine protein kinase, is encoded by all the members of the *Herpesviridae*, including human herpes simplex (HSV) HSV-1 and HSV-2 UL13 [1,2], varicella zoster virus (VZV) ORF47 [3], Epstein–Barr virus (EBV) BGLF4 [4,5], cytomegalovirus (CMV) UL97 [6,7,8], HHV-6 U69 [9], and Kaposi sarcoma-associated herpesvirus (KSHV) ORF36 [10]. The CHPK homologues are highly conserved, which suggests that CHPKs play important roles during virus replication and disease pathogenesis [1,11,12,13,14]. CHPKs have been implicated in several cellular and viral processes, including nuclear egress [15,16,17,18,19,20,21], tegument association/dissociation [22,23], viral gene expression [12,16,24,25,26], viral DNA replication [4,5,10,16,20,21,27,28,29], DNA damage responses [30,31], cell-cycle regulation [32,33,34,35], and the evasion of the interferon response [36,37,38]. Their importance is thought to be linked directly to their kinase activity, and several substrates have been identified [39,40]. While the specific substrates involved can vary between homologues, all CHPKs have been identified as part of the virion tegument [2,41,42], suggesting they play a role during the initiation of infection and virus entry.

We have shown that, in the case of *Gallid alphaherpesvirus* 2 (GaHV2), better known as Marek’s disease herpesvirus (MDV), CHPK and its kinase activity is required for transmission from chicken to chicken using the natural virus–animal host model [43,44]. This was shown by the mutation of the invariant lysine (K170) in MDV CHPK, which binds to ATP and is required for the kinase activity of CHPKs [5,8,10,11,19,45].

Poultry accounts for the type of meat with the second highest production in the world [46]. However, diseases such as Marek’s disease (MD) cause devastating rates of mortality and economic losses [47]. MDV produces immunosuppression, neurologic symptoms, metabolic dysregulation, and visceral organ tumors [48]. Natural infection begins by the host inhaling the virus, upon which cytolytic infection begins in the B cells and macrophages of the lung. From here, the virus disseminates into the lymphoid organs and travels to the skin. Once in the feather follicle epithelial (FFE) cells, cell-free virus is produced and shed into the environment. MDV CHPK is dispensable for in vitro cell replication, latency, and reactivation in T cells, but it is essential for horizontal transmission among chickens.

Historically, homologous non-oncogenic avian herpesviruses or attenuated MDV strains have been used to protect chickens against MD [49]. This includes other members of the *Mardivirus* genus, such as *Gallid alphaherpesvirus* 3 (GaHV3) and *Meleagrid alphaherpesvirus* 1 (MeHV1) or turkey herpesvirus (HVT), which have similar virus life cycles. These vaccines are effective in inhibiting clinical symptoms, especially tumor formation, but do not induce sterilizing immunity, which has resulted in increased virulence over the last five decades [50].

We hypothesize that 301B/1 CHPK is essential for the transmission of homologous avian herpesviruses, similarly to MDV. To test this hypothesis, we used an infectious bacterial artificial chromosome (BAC) clone of the MD vaccine GaHV3 strain 301B/1 that was fluorescently tagged in our laboratory [51]; additionally, a CHPK mutated virus was tested in experimental and natural infections in chickens. Our results conclusively show that 301B/1 CHPK is required for horizontal transmission and that rescuing the CHPK-null mutation restores transmission. These results support CHPK’s importance in horizontal transmission as a conserved function of *Herpesviridae*.

## 2. Materials and Methods

### 2.1. Cell Culture and Cells

Chicken embryo cells (CECs) were prepared from 10–11-day-old specific pathogen-free fertilized eggs obtained from the UIUC Poultry Farm (Urbana, IL, USA) using standard methods [52]. Primary CECs were seeded in a growth medium consisting of Medium 199 (Cellgro, Corning, NY, USA) supplemented with 10% tryptose-phosphate broth, 0.63% NaHCO_3_ solution, antibiotics, and 4% fetal bovine serum (FBS), and then reduced to 0.2% FBS when cells were confluent.

The chicken DF-1-Cre fibroblast cell line [53] was maintained in Leibovitz L-15 and McCoy 5A (LM) media (Gibco, Gaithersburg, MD, USA) supplemented with 10% FBS, antibiotics, and 50 µg/mL Zeocin (Invitrogen, Carlsbad, CA, USA).

All cells were maintained in a humidified atmosphere of 5% CO_2_ at 38 °C.

### 2.2. Generation of Recombinant (r) 301B/1

The parental virus in which 301B/1 expressing monomeric red fluorescent protein (mRFP) fused to pUL47 (pUL47mRFP) has been previously described [54]. The invariant lysine (K157) of 301B/1 was mutated to an alanine (A) to generate a CHPK mutant clone (rCHPKmut) using two-step Red-mediated mutagenesis in GS1783 *Escherichia coli* cells. Briefly, the I-*Sce*I-*aphAI* cassette from pEP-KanSII was amplified by PCR using Thermo Scientific Phusion Flash High-Fidelity PCR Master Mix (Thermo Fisher Scientific, Waltham, MA, USA) using the primers listed in Table 1. Subsequently, the K to A mutation was repaired back to K157 using the same approach. The primers used are listed in Table 1. Restriction fragment length polymorphism (RFLP) analysis, analytical PCR, and DNA sequencing using the primers listed in Table 2 confirmed all clones were correct.

r301Bs were reconstituted by transfecting DF-1-Cre cells with purified BAC DNA plus jetOPTIMUS^®^ (Polyplus, New York City, NY, USA) using the manufacturers’ instructions. After 2–3 days, transfected DF-1-Cre cells were mixed and seeded with fresh primary CECs until plaques formed, then further propagated in CECs until virus stocks could be stored. All r301Bs were used at ≤6 passages for cell culture and animal studies.

### 2.3. Immunofluorescence Assays (IFAs)

CEC cultures were infected with different r301B/1 viruses in 6-well tissue culture plates at 500 plaque-forming units (PFU) per well. At 5 days post-infection (dpi), cells were fixed with PFA buffer (2% paraformaldehyde, 0.1% Triton X-100) for 15 min and then washed twice with PBS. Cells were stained with anti-GaHV3 chicken sera and anti-GaHV3 gB (Y5.9) monoclonal antibody. Either goat anti-chicken IgY- or anti-mouse IgG-Alexa Fluor^®^ 488 secondary antibody (Molecular Probes, Eugene, OR, USA) was used as a secondary antibody for both primary antibodies. The virus plaques were observed using an EVOS^TM^ FL Cell Imaging System (Thermo Fisher Scientific, Waltham, MA, USA) and compiled using Adobe^®^ Photoshop^®^ version 21.0.1.

### 2.4. Measurement of Plaque Areas

Plaque areas were measured as previously described [55]. Immunofluorescence assays (IFAs) were performed as previously described [51] using anti-GaHV3 chicken sera. Digital images of 50 individual plaques were collected using an EVOS^TM^ FL Cell Imaging System (Thermo Fisher Scientific) and plaque areas were measured using ImageJ [56] version 1.53d software. Box-and-whisker plots were generated using Microsoft^®^ Excel^®^ for Microsoft 365, and significant differences were determined using the IBM^®^ SPSS^®^ Statistics version 28 software.

### 2.5. Viral Replication Kinetics in Cell Culture

To measure the replication of viruses in cell culture, qPCR assays were used to measure the relative level of replication, as previously described [55]. Briefly, CEC cultures were prepared in 6-well tissue culture plates and seeded with 10^3^ PFU/well of each virus. Total DNA was collected from the inoculum and at 48, 72, and 96 h following infection using DNA STAT-60^TM^ (Tel-Test, Inc., Friendswood, TX, USA) according to the manufacturer’s instructions. Quantification of 301B/1 genomic copies in CEC cultures was performed using primers and probe against 301B/1 gB (Probe: 5′-TexRed-tggctgcgtttctagcttactggt-IABlackRQ-3′; Forward: 5′-gagctttagcggtaggattgat-3′; Reverse: 5′-cgtcactggatatagggctttc-3′) and previously described chicken iNOS genes [55,57] obtained from Integrate DNA Technologies, Inc. (Coralville, IA, USA). All qPCR assays were performed in an Applied Biosystems QuantStudio 3 Real-Time PCR System (Thermo Fisher Scientific) and the results were analyzed using the QuantStudio^TM^ Design & Analysis Software version 1.4.2 supplied by the manufacturer. The fold-increase over inoculum was determined in triplicate for each virus and time point.

### 2.6. Animal Experiments

Commercial Pure Columbian chickens [58] were obtained from the UIUC Poultry Farm (Urbana, IL, USA). They were the offspring of MD-vaccinated parents and were therefore considered to be maternal antibody-positive. All experimental procedures were conducted in compliance with approved Institutional Animal Care and Use Committee protocols. Water and food were provided *ad libitum*.

In experiment 1, three-day-old chicks were inoculated by intra-abdominal inoculation with 10,000 PFU in 0.5 mL volumes of v301B47 or v3-CHPKmut, and housed in separate rooms (*n* = 10/group). For each group, another 9 naïve contact chickens were housed with infected birds to measure natural infection by horizontal transmission. Differences in the number of birds in the results are due to early chick mortalities unrelated to infection in the first week. In experiment 2, three-day-old chicks were experimentally infected by intra-abdominal inoculation with 10,000 PFU in 0.5 mL volumes of v301B47, v3-CHPKmut, or v3-CHPKmutR, and housed in separate rooms (*n* = 10/group) with age-matched, naïve contact chickens (*n* = 10/group). Experiments were terminated at 56 and 55 dpi for experiment 1 and 2, respectively.

### 2.7. Viral Replication Kinetics in Chickens

Whole blood was collected by wing-vein puncture at different time points for each group (*n* = 8/group) and DNA was extracted using the E.Z. 96 blood DNA kit from Omega Bio-tek, Inc. (Norcross, GA, USA) using the manufacturer’s instructions. Quantification of 301B/1 genomes in blood was performed exactly as described for viral replication kinetics in cell culture.

### 2.8. Monitoring v301B/1 in Feather Follicles (FFs)

To track the time at which each r301B47R or its derivatives reached the FFs, two flight feathers were plucked from the right and left wings (4 total) of inoculated birds weekly, and pUL47mRFP expression was examined using a Leica M205 FCA fluorescent stereomicroscope with a Leica DFC7000T digital color microscope camera (Leica Microsystems, Inc., Buffalo Grove, IL, USA).

### 2.9. Detection of Anti-301B/1 Antibodies in Sera of Chickens

To determine whether contact chickens were infected with v301B/1, sera were collected from all birds at termination and IFAs were performed as described above, except only fixed v301B47R-infected CECs were used and sera from contact chickens were diluted 1:10 in PBS. Goat anti-chicken IgY-Alexa Fluor^®^ 488 was used as a secondary antibody. Sera collected from experimentally infected chickens were used as positive controls. Plaques were observed using an EVOS^TM^ FL Cell Imaging System (Thermo Fisher Scientific, Waltham, MA, USA).

### 2.10. Statistical Analyses

Statistical analyses were performed using IBM SPSS Statistics version 28 software (SPSS Inc., Chicago, IL, USA). The significant differences for the plaque size assays were determined with Kruskal–Wallis tests (one-way non-parametric ANOVA), followed by multiple comparison tests. The normalized data of viral replication (qPCR) in blood and cell culture were analyzed using two-way ANOVA followed by Fisher’s LSD and Tukey’s post-hoc tests; virus (V) and time (T) and all possible interactions (V × T) were used as fixed effects, and the genomic copies were used as dependent variables. Fisher’s exact tests were used for infection and transmission experiments. Statistical significance was declared at *p*  <  0.05 and the mean tests associated with significant interactions (*p* < 0.05) were separated with Tukey’s tests.

## 3. Results

### 3.1. Generation and Characterization of Viruses

#### 3.1.1. Generation of r301B/1 with Mutated and Rescued CHPK

We previously developed a tool to track 301B/1 in cell culture and chickens [54]. We hypothesized that 301B/1 CHPK activity, like MDV CHPK activity, would be essential for horizontal transmission in chickens. Introducing a mutation in the invariant lysine of 301B/1 CHPK would destroy the kinase activity (Figure 1b); therefore, we mutated the lysine (K) at position 157 of 301B/1 CHPK to an alanine (A) to generate r3-CHPKmut (Figure 1c). To generate a rescued virus (r3-CHPKmutR), the alanine (A) at position 157 was mutated back to a lysine (K) (Figure 1d). RFLP analysis confirmed the integrity of the BAC clones, as the predicted banding pattern was observed (Figure 1c,d). In addition, DNA sequencing was used to confirm that each clone was correct at the nucleotide level (data not shown) using primers specific for the 301B/1 UL13(CHPK) gene (Table 2).

#### 3.1.2. Replication of r301B/1 with Mutated and Rescued CHPK in Cell Culture

Following the reconstitution of r301B/1 clones with CHPK-null mutation (r3-CHPKmut) and rescued CHPK (r3-CHPKmutR), we first tested replication in CEC cultures using plaque size and viral replication kinetic assays. Plaque size assays revealed a significant difference in the replication of v3-CHPKmut compared to v301B47R, while the rescuant virus (v3-CHPMmutR) was not different from v301B47R (Figure 2a). Viral replication kinetics showed no significant differences (*p* > 0.05) between the viruses at each time point (Figure 2b). Representative plaques for each virus are shown in Figure 2c,d. In all, these results suggests that 301B/1 CHPK plays a role in cell-to-cell spread, based on plaque size assays; however, it does not seem to affect overall viral DNA replication based on replication kinetics in cell culture.

### 3.2. GaHV3 CHPK Is Required for Horizontal Transmission in Chickens

#### 3.2.1. Mutation of GaHV3 K157 to A of CHPK Abrogates Transmission in Chickens

To test our hypothesis that GaHV3 CHPK is essential for horizontal transmission in chickens, we first tested our newly derived v3-CHPKmut using our experimental and natural infection model for horizontal transmission. To do this, chickens were inoculated with 10,000 PFU of v301B47R (*n* = 9) or v3-CHPKmut (*n* = 7) and housed with 8 and 6 uninfected contact chickens, respectively, over the course of 8 weeks. Using qPCR assays to measure 301B/1 replication in the blood of experimentally infected chickens (Figure 3a), no significant differences were observed in replication between the viruses (V), the time (T), and their interaction (V × T). However, when infection in FFs was monitored, there was a significant delay in positivity with v3-CHPKmut (Figure 3c) at 14 and 21 dpi (*p* < 0.05). There was no difference in the total percentage of experimentally infected birds, with 100% found to be positive by 42 dpi. When contact chickens were monitored for natural infection, no chickens housed with v3-CHPKmut-infected birds became infected compared to 88% of contact chickens housed with v301B47R. Following termination, whole blood was collected from all contact chickens and serum was tested for anti-GaHV-3 antibodies using IFA, as well as qPCR assays for 301B/1 viral DNA in the blood. It was confirmed that all chickens negative for fluorescent FFs were also negative for anti-GaHV-3 antibodies and 301B/1 viral DNA in the blood (data not shown).

#### 3.2.2. Repair of GaHV3 A157 to K of CHPK Restores Transmission in Chickens

Since v3-CHPKmut was unable to transmit to contact chickens, a second experiment was conducted including v301B47R, v3-CHPKmut, and v3-CHPKmutR. Here, 10 chickens were inoculated with 10,000 PFU of each virus and housed with 10 uninfected contact chickens over the course of 63 dpi. Using qPCR assays to measure 301B/1 replication in the blood of experimentally infected chickens (Figure 3b), two-way ANOVA revealed a significant difference between viruses (V; *p <* 0.05) and a tendency within DPI (D, *p =* 0.09) in replication, while differences were non-significant for the interaction between virus and DPI (V × D, *p >* 0.05). Overall, v301B47R showed a tendency (V, *p =* 0.06) to be different from v3-CHPKmut, while not being different (V, *p >* 0.05) from v3-CHPKmutR. There was no difference between v3-CHPKmut and v3-CHPKmutR, while v301B47R showed significantly higher viral loads at 21 and 29 dpi than both v3-CHPKmut and v3-CHPKmutR.

However, when contact chickens were monitored for natural infection, no chickens housed with v3-CHPKmut-infected birds became infected compared to 80% and 90% of contact chickens housed with v301B47R and v3-CHPKmutR, respectively (Figure 3d). Following termination, whole blood was collected from all chickens and blood was used to measure 301B/1 viral DNA. It was confirmed negative for all contact chickens housed with v3-CHPKmut (data not shown). With these results, we can conclude that 301B/1 CHPK is required for the horizontal transmission of the 301B/1 MD vaccine strain and that rescuing the CHPK mutation (A157K) restores its ability to transmit.

## 4. Discussion

In this study, we tested the importance of the *Herpesviridae* conserved CHPK for the horizontal transmission of the MD vaccine strain 301B/1. We were able to conclusively show that 301B/1 CHPK is essential for the horizontal transmission of 301B/1. These data reinforce the essential role of CHPK as a conserved function of avian herpesviruses. The exact role of CHPKs during horizontal transmission is not completely understood; however, its absolute requirement is most likely related to virion egress from the skin of infected chickens or the initiation and establishment of infection in the new host.

One result that was not expected was the inability of the CHPK rescuant virus (v3-CHPKmutR) to replicate like the wildtype (v301B47R) in experimentally infected chickens (Figure 3b,d). Normally, this would be a concern during animal studies; however, these results emphasized the importance of 301B/1 CHPK for natural infection. The fact that the CHPK mutant (v3-CHPKmut) and rescuant (v3-CHPKmutR) replicated at lower levels compared to the wild-type (v301B47R) in experimentally infected chickens shows that, even with a lower level of replication in chickens, the CHPK rescuant was able to infect contact chickens, while the CHPK mutant was still unable to spread (Figure 3d), emphasizing our interpretation that CHPK is required for transmission.

An important question we were unable to address in this report is the effect of the CHPK mutation (K157A) on the expression of 301B/1 CHPK, since we do not have an antibody against 301B/1 CHPK. In the case of MDV CHPK, it has been shown that the overall protein level of CHPK is diminished, which has been attributed to protein stability [59]. This was shown using epitope tagged MDV CHPK, and the same approach could be applied in the future to address this question. It is possible that part of the defect in transmission is due to a severe defect in 301B/1 CHPK expression.

Most MD vaccines do not transmit efficiently and fail to induce protective immunity, and, because of this, vaccines have not been able to compete with circulating virulent viruses. In fact, it has been experimentally shown that vaccination can enhance the transmission of virulent MDV in the field [60]. 301B/1 is as effective as traditional vaccine strains [61] and is able to spread among a chicken flock without disease signs [51]. This could lead to better protection for unvaccinated chickens and its replication could compete with virulent MDV in the skin and reduce the shedding of virulent MDV. However, this could also provide evolutionary pressure for virulent MDV to replicate and spread and result in MDV that spreads faster. Thus far, increasing the virulence of MDV over the last few decades has resulted in earlier morbidity and mortality, but has not resulted in the earlier shedding of infectious virus. The reason for this is not clear, but it could be due to minimal competition within FFE cells, which could provide vaccinal pressure to outcompete MD vaccines. Therefore, the question remains as to whether it would be beneficial for MD vaccines to efficiently transmit, and using wild-type GaHV3 strain 301B/1 and a 301B/1 CHPK mutant could be used to address this question.

In summary, our data extend the current knowledge of CHPK in the context of vaccinal spread and show that the importance of MDV CHPK for natural infection is not unique to MDV. Further studies are warranted to address CHPK’s role in natural infection in other herpesvirus–host models.

## Figures and Tables

**Figure 1 viruses-14-00586-f001:**
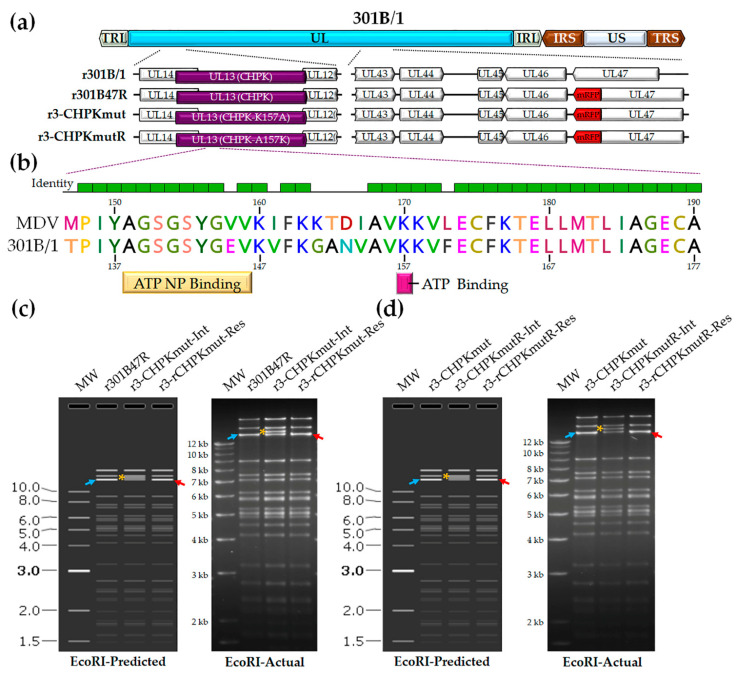
Generation of r301B/1clones. (**a**) Schematic representation of the 301B/1 infectious clone genome depicting the locations of the terminal repeat long (TRL) and short (TRS), internal repeat long (IRL) and short (IRS), and unique long (UL) and short (US) regions. The region of the UL spanning UL14 to UL12 and UL43 to UL47 are expanded to show the relevant genes within this region and modifications for each r301B/1 clone. (**b**) Alignment of MDV (RB-1B strain) and 301B/1 CHPKs at the ATP nucleotide binding (NP) and ATP binding sites using MUSCLE Alignment in Geneious Prime 2021.0.3 (Biomatters, Inc., San Diego, CA, USA). Lysine (K) at position 170 of MDV CHPK is required for the horizontal transmission of MDV [44] and the corresponding K157 of 301B/1 was targeted in this report. (**c**) The predicted (SnapGene software (from Insightful Science; available at snapgene.com) and actual RFLP analysis for generation of r3-CHPKmut (CHPK-K157A) of r301B/1 clones. BAC DNA obtained for r301B47R, the r3-CHPKmut-integrate (r3-CHPKmut-Int), and resolved clone (r3-CHPKmut-Res) were digested with EcoRI and electrophoresed through a 1.0% agarose gel. Integration of the *AphAI* sequence with the K157A mutation resulted in an increase in the 14,356 bp fragment (→) to 15,384 bp (*). Resolution of the *AphAI* sequence reduced the 15,384 bp fragment by 1028 bp to 14,356 bp (←). (**d**) The predicted and actual RFLP analysis for the generation of the r3-CHPKmut rescuant virus (r3-CHPKmutR) in which the mutated A157 was changed back to a lysine (CHPK-A157K). BAC DNA obtained for r3-CHPKmut, r3-CHPKmutR-Int, and r3-CHPKmutR-Res were digested with EcoRI and electrophoresed through a 1.0% agarose gel. Fragments generated were exactly as described for generation of r3-CHPKmut (**c**). The 1 kb Plus DNA Ladder from Invitrogen, Inc. (Carlsbad, CA, USA) was used as a molecular weight marker. No extraneous alterations are evident.

**Figure 2 viruses-14-00586-f002:**
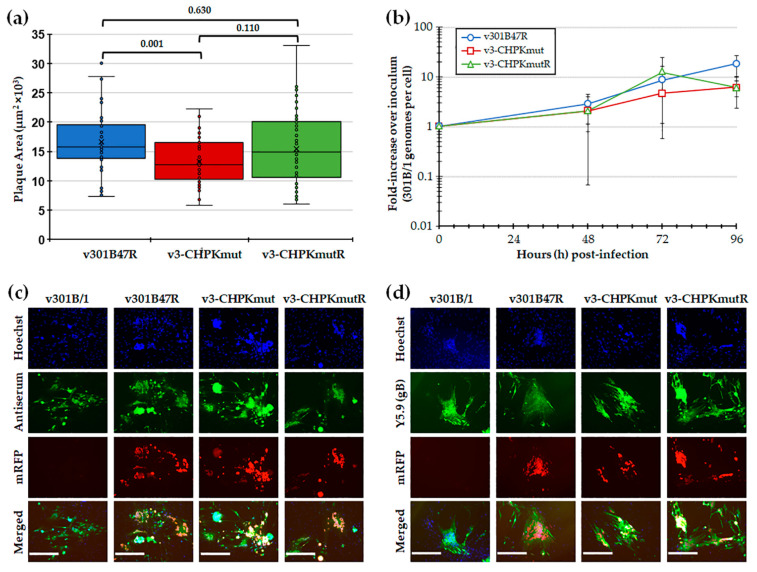
Replication in tissue culture cells. (**a**) Mean plaque areas (*n* = 50) of viruses reconstituted from r301B47R, v3-CHPKmut, and v3-CHPKmutR were measured and the results are shown as box-and-whisker plots. Statistically significant differences between the different viruses were determined with Kruskal–Wallis tests (one-way non-parametric ANOVA), followed by multiple comparison tests. There was a significant difference (*p =* 0.001) between v301B47R and v-3CHPKmut using an independent samples Kruskal–Wallis test. No significant difference (*p =* 0.630) was determined between v301B47R, v-3CHPKmut, and v-3CHPKmutR. (**b**) Multi-step replication kinetics was used to measure virus replication in CEC cultures. The mean fold-change in viral DNA copies over the inoculum is shown for each virus and time point. Except for an effect of significant changes in viral DNA replication over time (T; *p =* 0.001), there were no differences in virus replication between viruses (V; *p =* 0.308) or in their interaction (V × T; *p =* 0.19) (*p >* 0.05, two-way ANOVA, Tukey, *n* = 6). (**c**) Representative plaques for v301B/1, v301B47R, v3-CHPKmut, and v3-CHPKmutR used for plaque size assays are shown. Plaques were stained with polyclonal chicken anti-GaHV-3 antibody and goat anti-chicken-IgY Alexa488 (green) was used as a secondary antibody to identify plaques. Fluorescent expression of mRFP (red) was directly visualized and cells were counterstained with Hoechst 33,342 to visualize nuclei. (**d**) Representative plaques for v301B/1, v301B47R, v3-CHPKmut, and v3-CHPKmutR were stained with monoclonal mouse anti-gB IgG1 antibody (Y5.9) and goat anti-mouse-IgG Alexa488 (green) was used as a secondary antibody to show 301B/1 specificity of plaques. Fluorescent expression of mRFP (red) was directly visualized and cells were counterstained with Hoechst 33,342 to visualize nuclei. Scale bars represent 200 µm.

**Figure 3 viruses-14-00586-f003:**
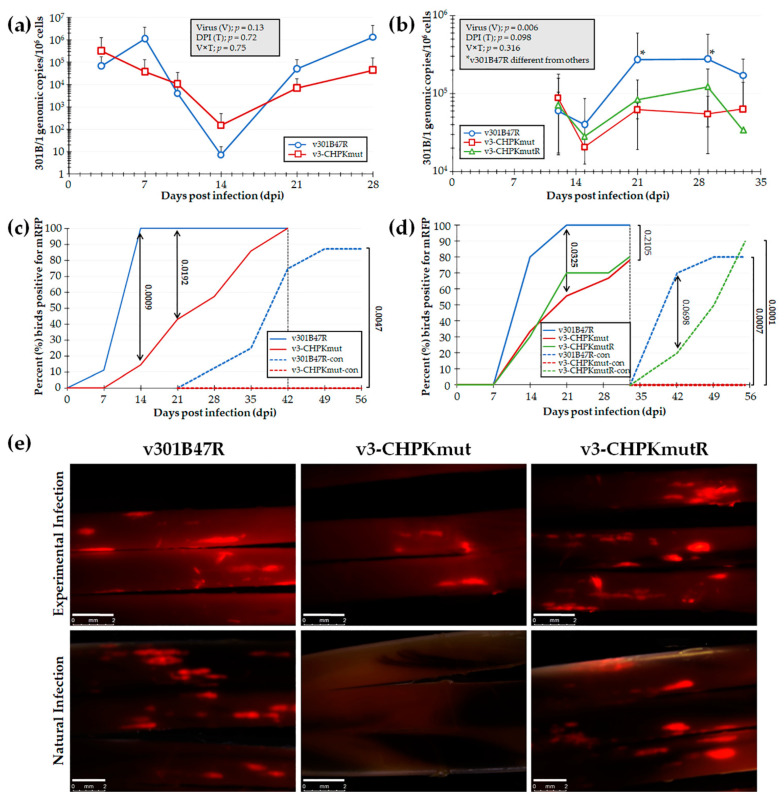
Replication and interindividual spread of r301B/1 viruses in chickens. Pure Columbian chickens were experimentally infected with v301B47R, v-3CHPKmut, or v3-CHPKmutR, as described in the Materials and Methods, in two experiments. (**a**,**b**) Replication was monitored in experimentally infected chickens by the quantification of 301B/1 genomes in blood over the first 4 (**a**) and 5 (**b**) weeks of infection. The mean 301B/1 genomic copies per 10^6^ blood cells ± standard error of means are shown. Two-way ANOVAs were performed to check the significant effect of virus and time (dpi) and the associated interaction effect (V × D). (**a**) No significant differences (*p* > 0.05, *n* = 109) were determined between both viruses. (**b**) Significant differences were seen (*p* < 0.05, *n* = 75) between v301B47R and both v3-CHPKmut and v3-CHPKmutR at 21 and 29 dpi (*). Samples collected at 4 and 7 dpi lacked DNA and were excluded from data analysis. (**c**,**d**) Quantitative analysis of the percent of birds positive for pUL47mRFP in FFs over the course of both experiments. Using Fisher’s exact test at *p* < 0.05, there were no differences in the total percentage of experimentally infected birds infected at 42 (**c**) and 33 (**d**) dpi. No contact chickens housed with v3-CHPKmut-infected birds became infected in both experiments. (**e**) Representative feather follicles plucked from chickens experimentally and naturally infected with each virus in the second experiment. The white line represents a 2 mm scale bar.

**Table 1 viruses-14-00586-t001:** Primers used for the generation of recombinant 301B/1.

Modification ^1^	Direction ^2^	Sequence (5′→3′) ^3^
UL13-K157A	Forward	AGCTATGGAGAAGTTAAAGTATTTAAGGGTGCAAATGTAGCCGTCGCGAAGGTGTTCGAGTGTTT*TAGGGATAACAGGGTAATCGATTT*
	Reverse	CAGTGTCATAAGCAATTCGGTCTTGAAACACTCGAACACCTTCGCGACGGCTACATTTGCACCCT*GCCAGTGTTACAACCAATTAACC*
UL13-A157K	Forward	AGCTATGGAGAAGTTAAAGTATTTAAGGGTGCAAATGTAGCCGTCAAAAAGGTGTTCGAGTGTTT*TAGGGATAACAGGGTAATCGATTT*
	Reverse	CAGTGTCATAAGCAATTCGGTCTTGAAACACTCGAACACCTTTTTGACGGCTACATTTGCACCCTGCCAGTGTTACAACCAATTAACC

^1^ Modification of the 301B/1 genome using two-step Red recombination. ^2^ Directionality of the primer. ^3^ *Italics indicate the template-binding region of the primers for PCR amplification with pEP-KanSII*. Red indicates unique upstream integration sequences. Green indicates unique downstream integration sequences. Nucleotides targeted (position 157 of UL13) for mutagenesis are underlined.

**Table 2 viruses-14-00586-t002:** Primers used for sequencing.

Gene ^1^	Direction ^2^	Sequence (5′→3′)
301B/1 UL13	Forward	GCGATCGCCTTCCAGACATA
	Reverse	AAGGTTTGGACTGCACTGCT
	Forward	CGCCAATATATGCGGGAAGC
	Reverse	CGATGGCAGTACGAGTCCAT

^1^ Gene sequenced with the set of primers. ^2^ Directionality of the primer.

## Data Availability

Not applicable.

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
