# Peer review of "The Conserved Herpesviridae Protein Kinase (CHPK) of Gallid alphaherpesvirus 3 (GaHV3) Is Required for Horizontal Spread and Natural Infection in Chickens"

_viruses, 2022, doi:10.3390/v14030586_

Round 1

Reviewer 1 Report

This article („The Conserved Herpesviridae Protein Kinase (CHPK) of Gallid Alphaherpesvirus 3 (GaHV3) is Required for Horizontal Spread and Natural Infection in Chickens” by Krieter A et al.) could be seen as a follow up study to an earlier publication by some of the same authors in JVI https://doi.org/10.1128/JVI.01522-19. The work aims at deciphering the role of the multifactorial conserved herpesvirus protein kinase (CHPK), which might possess conserved functions in GaHV-2 and -3 and seems essential for efficient horizontal virus spread in GaHV3 infection.

However, some parts of the manuscript overlap quite a bit with the abovementioned JVI manuscript and I suggest you substantially edit this text to avoid plagiarism (compare the first intro paragraphs, for example…).

Below, I list some comments that need to be addressed during preparation of a revised version of the manuscript. So far, I can only recommend to reject this manuscript but are happy to review a substantially revised version that includes more data and a more thorough discussion including a discussion of the limitations of this work.

References are missing in lines 47 – 54.

Table 2: Are all four primers targeting UL13? And if yes, why did you use two different primer sets? Please clarify (also see line 183).

Lines 104f: On which day p.i. were the plaques stained and analysed?

Lines 122f: Please include information on the viral and host genes that were analysed by qPCR.

Lines 134f: Could you include information on the MHC haplotype(s) of the commercial Pure Columbian chickens used? Why were the chickens infected with 10,000 pfu (which is a very high infectious dose)? When were the experiments terminated? At 56 dpi? Were all experiments performed in a blinded fashion? What was the injection volume? How many birds were sampled at each time point?

Figure 1: I think it could be helpful to provide more information on the ATP NP and ATP binding sites in the introduction of this manuscript. Why are these sites important, what exactly happens upon mutation of K157 (or, K170 in GaHV2)? How do you explain the extra band at roughly 8kb in the wt BAC in panel d (and maybe also in panel c as a very faint band, if you play with the saturation)?

Figure 2: You show plaques that were stained with antisera and with Y5.9. Yet, you never explain why you chose to show both stainings. Please add some information on this. Do you have syncytia formation only with the mutants and not with the wt? The plaques appear to be pretty amorphous – how reliable are the manual plaque size measurements with this virus? Were these analyses performed in a blinded fashion? The DAPI staining is so much brighter in the wt in panel d (see merged image). I suggest you show representative images that are comparable regarding the fluorescence intensities. The scale bar labels are not readable and should be increased in size or removed.

Figure 3: Why do you only show the data up to ~32 dpi for the experimentally infected chickens in panel d? Please show these missing data! Please also include the qPCR data for the earlier time points in panel b of this figure. What does n=109 and n=75 indicate in the figure legends? Certainly not the amount of chickens that you sampled?

Line 218: I think these are typos, it should read “from r301B47R, r3-CHPKmut, and r3-CHPKmutR”… Correct?

Do you have any data on the expression of your mutated CHPK? Any speculations if it is expressed as efficiently as the wt?

Results (starting from line 236):

  1. How was the serum was tested for anti-GaHV-3 antibodies using IFA (line 251)? This procedure should be included in the M&M.
  2. Instead of “data not shown” in lines 254 and 272, you should include the 56 dpi time point in Figure 3, panels a and b.
  3. How do you explain the big differences between the wt v301B47R and v3-CHPKmut in experiment 1 and 2 regarding MDV genome copies in blood samples? How do you explain the huge differences in viral titers between the two experiments (panel a vs b in Fig. 3)?
  4. Why does the revertant virus not fully restore the wt phenotype? You even report statistically significant differences in line 266 which render the revertant control invalid in this assessment. The feather pictures in Fig 3, panel e are not representative as well.
  5. In addition, the legend to panel b states that the asterisks indicate “v301B47R different from others”. So your revertant really seems somehow screwed. I suggest you NGS-seq all three viruses and look for off target mutations, or how do you explain these differences?

Line 281: Please correct “Two-ANOVA”.

Lines 296f are redundant and should be deleted (see lines 35f). Instead, you should prepare a more thorough discussion of the presented work. Lines 304 – 314 are entirely discussing issues that seem detached from what you want to convey with their story. Is the goal here to suggest that 301B/1 should be used as a commercial vaccine? And how does that relate to the presented data? How can differences to GaHV2 be explained since in GaHV2 infections, CHPK is not required to efficiently infect FFE cells in vivo?

This is a very short manuscript with i) a short intro ii) only 3 figures and iii) a short and rather superficial and incomplete discussion. I suggest you complete the manuscript and maybe resubmit it as a short communication. Using that article format, you could include the bit of discussion into the respective results paragraphs and conclude this manuscript with a short and crisp conclusion paragraph.

Reviewer 2 Report

In this article, the authors mutated the ATP-binding domain of the conserved herpesvirus protein kinase (CHPK) of the Marek's Disease Virus vaccine strain 301B/1 and sought to determine its effects on replication and spread.  Importance of this kinase has been described with other herpesviruses including MDV.

The major findings are that plaque size is reduced compared to the parental strain, however it is not significantly different from the revertant rescue virus.  While the kinase mutant could replicate comparably in culture, it was delayed in infected chickens, however again equally impaired as the revertant virus.  While the poor performance of the revertant in these two assays precludes conclusive evidence that this one mutation is to blame for these phenotypes, the strongest phenotype is that the kinase mutant was entirely incapable of spreading to co-housed animals while the parental and revertant virus spread well.

Over all, the methodology and conclusions are solid and the text well written.  In my opinion this article can be published as it currently is.

Author Response

Thank you for your review.

Reviewer 3 Report

Important finding and well written manuscript. 

Author Response

Thank you for your review.

Reviewer 4 Report

The article entitled “The Conserved Herpesviridae Protein Kinase (CHPK) of Gallid Alphaherpesvirus 3 (GaHV3) is Required for Horizontal Spread and Natural Infection in Chickens” presents an accurate molecular comparative virology study. The manuscript is well written. I found only one typing mistake in line 214: “assasys” instead of “assays”. I have only one comment on the manuscript, namely that only Figure 1/B in the article shows that a K157A mutation in 301B/1 CHPK disrupts the ATP binding site, thereby inactivating the kinase. It would be very useful to write a few sentences about this fact at the end of the introduction. So, I recommend this MS for publication in Viruses with the aforementioned minor revision.

Author Response

Thank you for your review. We have fixed the misspelling of "assays" on line 214 (original).  Also, to address the other question, we have included the following sentence in the Introduction, “This was shown by mutation of the invariant lysine (K170) in MDV CHPK that binds to ATP and is required for kinase activity of CHPKs [5,8,10,11,19,45]. Lines 46-48.

Round 2

Reviewer 1 Report

I appreciate the thorough response of the authors to my comments and concerns. The revisions they have made have improved the paper in my opinion and I only have five additional comments.

  1. I suggest that the authors revise lines 75-88 of this text to avoid the very obvious Ctrl-C/Ctrl-V from “Expression of the Conserved Herpesvirus Protein Kinase (CHPK) of Marek’s Disease Alphaherpesvirus in the Skin Reveals a Mechanistic Importance for CHPK during Interindividual Spread in Chickens”.
  2. Footnote 3 is missing in Tab. 2.
  3. The previous issue remains and should be briefly discussed by the authors: How do the authors explain the big differences between the wt v301B47R and v3-CHPKmut in experiment 1 and 2 regarding genome copies in blood samples? How can the huge differences in viral titers between the two experiments be explained (panel a vs b in Fig. 3)?
  4. Is there a good reason why the authors used different magnifications in Fig. 3e (apparent because of different lengths of the scale bars)?
  5. The authors write that “Thus far, increasing virulence of MDV over the last few decades has resulted in higher morbidity and mortality, but has not resulted in significantly increased shedding rates” (lines 341f) This has been recently questioned, however, by 10.1371/journal.ppat.1009104 where the authors show that meq signature mutations that are indicative of the respective pathotype seem to contribute to enhance virus shedding. It would be good to clarify that.

Author Response

1. I suggest that the authors revise lines 75-88 of this text to avoid the very obvious Ctrl-C/Ctrl-V from “Expression of the Conserved Herpesvirus Protein Kinase (CHPK) of Marek’s Disease Alphaherpesvirus in the Skin Reveals a Mechanistic Importance for CHPK during Interindividual Spread in Chickens”.

This has been rewritten as requested:

Chicken embryo cells (CECs) were prepared from 10-11-day-old specific pathogen-free fertilized eggs obtained from the UIUC Poultry Farm (Urbana, IL) using standard methods [52]. Primary CECs were seeded in growth medium consisting of Medium 199 (Cellgro, Corning, NY, USA) supplemented with 10% tryptose-phosphate broth, 0.63% NaHCO3 solution, antibiotics, and 4% fetal bovine serum (FBS) and then reduced to 0.2% FBS when cells were confluent. 

The chicken DF-1-Cre fibroblast cell line [53] was maintained in  Leibovitz L-15 and McCoy 5A (LM) media (Gibco, Gaithersburg, MD, USA) supplemented with 10% FBS, antibiotics, and 50 µg/ml Zeocin (Invitrogen, Carlsbad, CA).”

2. Footnote 3 is missing in Tab. 2.

Thank you for noticing this. There should not be a 3rd footnote and it has been removed.

3. The previous issue remains and should be briefly discussed by the authors: How do the authors explain the big differences between the wt v301B47R and v3-CHPKmut in experiment 1 and 2 regarding genome copies in blood samples? How can the huge differences in viral titers between the two experiments be explained (panel a vs b in Fig. 3)?

Thank you for noticing this.  When looking at the raw data, we found that the 301B/1 genomes were divided by the chicken genomic copies incorrectly.  The figure in Fig. 3b was calculated at 301B/1 genomic copies/103 cells and not 106 cells as is shown in Fig. 3a.  We have corrected this so both are /106 cells. We also noticed the figures had “MDV” on the Y-axis and has been corrected to “301B/1”.

4. Is there a good reason why the authors used different magnifications in Fig. 3e (apparent because of different lengths of the scale bars)?

Typically, we try to get as many feathers on the image as we can with the highest magnification for clarity. Hence, some are at higher magnifications as others.

5. The authors write that “Thus far, increasing virulence of MDV over the last few decades has resulted in higher morbidity and mortality, but has not resulted in significantly increased shedding rates” (lines 341f) This has been recently questioned, however, by 10.1371/journal.ppat.1009104 where the authors show that meq signature mutations that are indicative of the respective pathotype seem to contribute to enhance virus shedding. It would be good to clarify that.

Thank you for requesting clarity in this statement.  As far as we are aware, only viral loads were enhanced by specific Meq mutations in this report and the time to reach the feathers was not affected by the Meq mutations (10 days pi).  Using the term “rate” was not the correct term for making our point that the time to reach the feathers and produce infectious virus has not “decreased." Thus, shedding earlier has not occurred. We have restated this to be more specific as, “Thus far, increasing virulence of MDV over the last few decades has resulted in earlier morbidity and mortality, but has not resulted in earlier shedding of infectious virus. The reason for this is not clear but could be due to minimal competition within FFE cells that could provide vaccinal pressure to outcompete MD vaccines.